:ᄋ: PLOS | ONE

# Characterization of black patina from the Tiber River embankments using Next-Generation Sequencing

Federica Antonelli[1]*, Alfonso Esposito[2], Ludovica Calvo[3], Valerio Licursi[4], Philippe Tisseyre[5], Sandra Ricci[6], Manuela Romagnoli[1], Silvano Piazza[2], Francesca Guerrieri [3,7]*

1 Department of Innovation of Biological Systems, Food and Forestry (DIBAF), Tuscia University, Viterbo, Italy, 2 Department of Cellular, Computational and Integrative Biology–CIBIO, University of Trento, Trento, Italy, 3 Center for Life NanoScience@Sapienza, Istituto Italiano di Tecnologia, Rome, Italy, 4 Institute for Systems Analysis and Computer Science "Antonio Ruberti", National Research Council, Rome, Italy, 5 Soprintendenza del Mare, Regione Sicilia, Palermo, Italy, 6 Biology Laboratory, Istituto Superiore per la Conservazione e per il Restauro (ISCR), Rome, Italy, 7 Epigenetics and epigenomic of hepatocellular carcinoma, U1052, Cancer Research Center of Lyon (CRCL), Lyon, France

* fraguerrieri@gmail.com (FG); fedantonelli@gmail.com (FA)

**Data Availability Statement:** All data are available from the NCBI database (accession number PRJNA553109)

## Abstract

Black patinas are very common biological deterioration phenomena on lapideous artworks in outdoor environments. These substrates, exposed to sunlight, and atmospheric and environmental agents (i.e. wind and temperature changes), represent extreme environments that can only be colonized by highly versatile and adaptable microorganisms. Black patinas comprise a wide variety of microorganisms, but the morphological plasticity of most of these microorganisms hinders their identification by optical microscopy. This study used Next-Generation Sequencing (NGS) (including shotgun and amplicon sequencing) to characterize the black patina of the travertine embankments (*muraglioni*) of the Tiber River in Rome (Italy). Overall, the sequencing highlighted the rich diversity of bacterial and fungal communities and allowed the identification of more than one hundred taxa. NGS confirmed the relevance of coccoid and filamentous cyanobacteria observed by optical microscopy and revealed an informative landscape of the fungal community underlining the presence of microcolonial fungi and phylloplane yeasts. For the first time high-throughput sequencing allowed the exploration of the expansive diversity of bacteria in black patina, which has so far been overlooked in routine analyses. Furthermore, the identification of euendolithic microorganisms and weathering agents underlines the biodegradative role of black patina, which has often been underestimated. Therefore, the use of NGS to characterize black patinas could be useful in choosing appropriate conservation treatments and in the monitoring of stone colonization after the restoration interventions.

## Introduction

In the field of cultural heritage, the term "patina" has several meanings: the time-dependent darkening of frescos and oil paintings, the superficial oxidation of bronze and copper, and,

**Funding:** The authors received no specific funding for this work.

**Competing interests:** The authors have declared that no competing interests exist.

more generally speaking, the surface transformations that lead to the ageing of artworks. Since the late 1990s, this term has also been used to define an aesthetic change of rock surfaces linked to biological colonization [1]. The growth of microorganisms (bacteria, cyanobacteria, algae, fungi, and lichens) on lapideous surfaces, as well as aesthetic alteration, can cause an actual deterioration of the stone. The damage is predominantly linked to the production of organic and inorganic acids and to an euendolithic living habitus [2–6].

Black crusts, defined as "crusts developing generally on areas protected against direct rainfall or water runoff in urban environments [. . .], composed mainly of particles from the atmosphere trapped in gypsum ($CaSO_4.2H_2O$)" [7], are different from black patinas, which are biofilms composed of pigmented microorganisms and represent a very common deterioration phenomenon on lapideous artworks in outdoor environments. Lapideous artworks can be considered as extreme environments, characterized by inhospitable surfaces exposed to several stresses such as high solar radiation, desiccation and rehydration, considerable diurnal and annual temperature fluctuations, and lack of nutrients [8]. Consequently, they can only be colonized by microorganisms characterized by constitutive or fast adaptive cellular or metabolic responses to these conditions. The main adaptations are the production of UV-screening compounds, exopolymeric substances that retain an adequate water content, and constituent compounds of thick cell walls that protect cells against physicochemical hazards [8–13]. Black patinas contain microbial communities composed of a wide variety of microorganisms (mainly cyanobacteria, microalgae and rock inhabiting fungi (RIF) [8,11]) in different physiological states, that can live as either epiliths on the rock surfaces or as endoliths (cryptoendoliths, chasmoendoliths or euendoliths) within the substrate [4,6,14,15].

Phototrophic microorganisms are the first colonizers of rock surfaces in outdoor environments [16]. Their spores, cells, and propagation structures, dispersed by wind, water, and animals (such as birds, bats, and squirrels) [17], adhere to the rock and initiate biofilm formation in which a heterogeneous matrix of microorganisms is held together and tightly bound to underlying surfaces by extracellular polymeric substances (EPS) [8]. The growth of cyanobacteria on natural rocks leads to variously colored strips, known as Tintenstriche [18], the composition of which has been widely studied [for a review see 15]. Similarly, these patinas can be found on stone monuments exposed outdoors; Albertano (2012) [19] reports an exhaustive review of the works carried out on this topic. Phototrophic colonizers enrich substrates with organic carbon (produced during photosynthesis) and nitrogen (produced during nitrogen fixation by cyanobacteria) [20]. Consequently, the presence of these nutrients favors the settlement of heterotrophic microorganisms. Previous studies of RIF showed that these heterotrophic microorganisms can be divided into two groups: Hyphomycetes of soil and/or epiphytic origin, and microcolonial fungi (MCF) [8,21,22]. Since 2012, the class of Hyphomycetes has undergone a profound modification and numerous genera and species previously belonging to this class have been reassigned to other classes of Ascomycetes. MCF are ubiquitous colonizers of rock surfaces worldwide and possess particular features such as a very plastic morphology, a high degree of melanization, meristematic growth, and the abundant production of EPS [23]. They are Ascomycetes of the classes Dothideomycetes and Eurotiomycetes. Molecular phylogenetic studies classified these microorganisms as follow: Dothideomycetes RIF belong to the orders Capnodiales, Dothideales, and Pleosporales; while Eurotiomycetes RIF cluster in the lineages of Chaetothyriales and Verrucariales [22,24–26].

Several studies have demonstrated that chemoorganotrophic and chemolithotrophic bacteria form part of the microbial communities present on different types of rock [7]. From early 2000, the application of molecular techniques has confirmed the presence of Proteobacteria, Actinobacteria, Acidobacteria, and the Cytophaga–Flavobacterium–Bacteroides group in

subaerial biofilms present on stone artworks [27–29]; however, the importance of these micro-organisms and their biodegradative role has not yet been thoroughly investigated.

Knowledge of the taxa present in patinas, and of their ecological requirements, is necessary when choosing appropriate conservation treatments. It is, however, not always easy to identify the constituents of these microbial communities; due to their morphological plasticity, the identification of patina-related microorganisms by optical microscope observation is usually very difficult or not possible, and culturing methods are always limited by the special living conditions of these microorganisms [3]. By the end of the twentieth century, the introduction of molecular identification methods had contributed to the identification of a large number of microbial components of patinas that were previously unknown [14,30–32]; nevertheless, identifying the exact composition of a black patina remains challenging. Considering this, the first aim of the present study was to test high-throughput sequencing with Illumina platforms (Miseq and Nextseq) for the characterization of the black patina of the travertine embankments (*muraglioni*) of the Tiber River in Rome (Italy). The second aim of the study was to evaluate the efficacy of this molecular technique by comparing it to a traditional method routinely used for the characterization of biological colonization of artworks.

## Materials and methods

### Sample collection and description

The Tiber River's embankments were built between 1875 and 1926 on the basis of the project of the engineer Raffaele Canevari. They are riverbank walls made of travertine, more than 18 m high and 8 km long, and function in protecting the city from flooding. The embankments were chosen for this study on the biological colonization because they represent good examples of lapideous artwork that have been exposed in an urban outdoor environment for a long time.

Although no data are available on the origin of the travertine used in the river Tiber's embankments, it is likely that it was obtained from the outcrops of the thermal springs (Acque Albule) located at the foothills of Tiburtini Mountains, near Tivoli (Rome). In general, calcite represents over 99% of travertine composition, the remaining part is made up of layers rich in anhydrite, magnesite, quartz, sanidine, piroxenes, micas, garnets, and spinels [33]. The rock texture may be vacuolar or radiating fibrous with a high variability in terms of quantity, dimension, and shape of the pores. The number of pores and cavities in a slab, which depends on the slab's position in the deposit and on the cutting direction relative to the deposition of layers, highly influences the amount of water retained and bioreceptivity of the stone [34].

The city of Rome is characterized by a mid-latitude temperate climate, with hot summers and mild and relatively moist winters. The autumn is usually the rainiest season, while the incidence of snowfall is negligible. Information about rainfall trends and temperature, provided by a study carried at 21 stations over the period 1984–2014, show that in the urban area of Rome the average annual precipitation is 793 mm (784.4 mm registered from the Ostiense station, nearest to the sampling site) and the annual mean temperature is 14°C [35].

The patina and control samples were collected from the northwestern embankment of the Tiber River (Fig 1), along Porto di Ripa Grande, at a height of around 170 cm. For each sample 1 sq.cm of travertine was scraped with a sterile scalpel and the resulting powder preserved at 4°C. The black patina was randomly sampled from areas where no other biological colonization was present (i.e. mosses, lichens, plants). The areas covered with graffiti were avoided. The controls were collected from white, apparently un-colonized areas. Twenty samples of black patina (B) and twenty controls (uncolonized regions, U) were used for NGS (S1 Table), while three black patina and three controls samples were observed as unstained wet-mounts through an optical microscope (Leica DM RB).

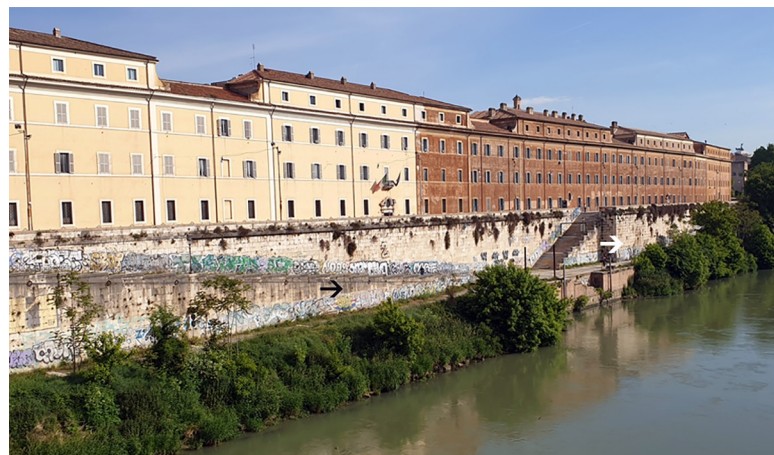

**Fig 1. The northwestern embankment of the Tiber River along Porto di Ripa Grande.** The arrows indicate two examples of sampled areas (black arrow = B; white arrow = U).

## DNA extraction

Each B and U powder was resuspended in 60 μl of lysozyme (10 mg/ml) and 240 μl of TE 1X. After vortexing, each sample was incubated for 30 min at 37°C. Subsequently, 300 μl of lysis buffer (#MC5001C Promega) and 30 μl of Proteinase K (#MC500C) were added to the samples, which were then incubated 20 min at 56°C. Total genomic DNA was extracted using a Maxwell® RSC Instrument (Promega, Wisconsin, USA) and a genomic DNA extraction kit (Promega, cat no. #AS1400), as per manufacturers' instructions. The total DNA extracted is reported in S1 Table.

## Library preparation and sequencing

The V3-V4 region of the 16S rRNA gene (amplified using the primers described in Illumina 16s protocol: # 15044223 Rev. B) and the ITS2 fungal region (amplified using the following primers: ITS3 PCR Forward Primer 5ʹ TCGTCGGCAGCGTCAGATGTGTATAAGAGACAG–G CATCGATGAAGAACGCAGC–3ʹ and ITS4 Reverse Primer 5ʹ GTCTCGTGGGCTCGGAGATG TGTATAAGAGACAG–TCCTCCGCTTATTGATATGC–3ʹ) were subject to amplicon library preparation (according to Illumina's instructions, 16S Metagenomic Sequencing Library Preparation, Part # 15044223 Rev. B). Shotgun libraries were prepared using Nextera XT library Prep (Illumina Cat. FC-131-1024). Eeach library was determined using Agilent 2200 Tapestation (Agilent Technologies, Santa Clara, CA, United States) and quantified using a Qubit 2.0 fluorometer with a Qubit dsDNA HS Assay Kit (cat# Q32851, Thermo Fisher Scientific, MA, United States). Sequencing was performed at the CLNS@Sapienza Genomics facitlity (Center for Life NanoScience@Sapienza, Istituto Italiano di Tecnologia, Rome, Italy), using Miseq (2x300 paired-end, 600-cycle) and Nextseq500 (2x150 paired-end, 300-cycle) Illumina platforms.

## Sequencing data analysis

Marker data were analyzed using QIIME2 (https://qiime2.org), according to the standard pipelines. Briefly, quality trimming and OTU-picking was done using DADA2 [36], representative sequences were aligned using MAFFT [37], uninformative positions were masked and a phylogenetic tree was built with FastTree [38]. The alpha diversity values and beta diversity

(i.e. UniFrac distance) were calculated on rarefied samples. Rarefaction values (3000 reads for 16S and 300 for ITS) were chosen upon observation of rarefaction curves (S1 Fig). Assessment of significant variation of alpha diversity between categories was determined using the Kruskal-Wallis test. Beta diversity significance (among categories) test was calculated with PERMANOVA and Mantel test, respectively. Taxonomic assignments were made for representative sequences using the most updated version of the SILVA database (release 132) [39], or the UNITE database (for fungal data) [40]. The feature classifier was trained using the QIIME2 classify-sklearn plugin on the database; the same plugin was used to classify the reads in the real dataset. All the unassigned sequences were further inspected using BLAST on the National Center for Biotechnology Information's (NCBI) 16S ribosomal RNA sequences database to integrate the taxonomic assignment.

Raw shotgun sequencing reads were co-assembled using MEGAHIT [41], the resulting contigs were imported into the advanced analysis and visualization platform, Anvi'o, for subsequent analysis [42]. Reads were re-mapped on the contigs to obtain coverage information, and contigs were binned according to their k-mer frequency and coverage using CONCOCT [43]. We only retained genomes with a completion score above 90% (calculated as the percentage of single-copy genes retrieved in the genome) and redundancy below 10%. A putative taxonomy was assigned to the bins using PhyloPhlAn [44], considering the Average Nucleotide Identity (ANI) of the genome bin with the closest relative found by the program. Functions encoded in the MAGs were classified according to the COG categories using eggNOG-mapper [45].

## Results

### Microscope observations

Light microscope (Leica DM RB) observations of black patina samples revealed the presence of bacteria, cyanobacteria, chlorophyta and, fungi. Bacteria and cyanobacteria were the most abundant taxa in all the observed wet-mounts. Cyanobacteria were either coccoid or filamentous forms, present as either single cells or colonies. The most abundant coccoid cyanobacteria were spherical cells, rarely solitary but were most frequently grouped in irregular agglomerations or formed roughly spherical colonies. The cells had a green or yellowish content and a thin yellowish or brownish sheath. Other sub-spherical cells, 6–18 μm in diameter, green-yellowish in color, and mainly grouped in dyad arrangements or forming small cell aggregates were identified as *Chroococcus lithophilus* Ercegovic 1925 (Fig 2A). The filamentous cyanobacteria (12–20 μm wide) were unbranched, with cells 8–16 μm wide, distinctly shorter than wide or at most as long as wide, and covered by a thin colorless of slightly yellowish sheath. These cells were thought to be part of the family Scytonemataceae (order Scytonematales). Coccoid green algae, 12–19 μm in diameter were also observed within these samples (Fig 2B). The cells were spherical, relatively thin walled with a central chloroplast and one pyrenoid. Fungal structures (spores and hyphae) were also present in all the samples. The hyphae were composed of spherical or slightly elongated cells, 5–10 μm in diameter, with thick melanized walls typical of black meristematic fungi (Fig 2C). The microbial communities of control samples were composed mainly of bacteria with few, single cells of coccoid cyanobacteria. With the exception of *C. lithophilus*, it was not possible to identify the other observed microorganisms due to the lack of taxonomical features.

### Bacterial community

The Illumina Miseq platform was used to sequence the bacterial 16S rRNAs V3-V4 region of thirty-two samples (12 from uncolonized controls and 20 from black patina). Only 12 of the 20 control samples yielded detectable amplicon levels; no amplification was detected in the

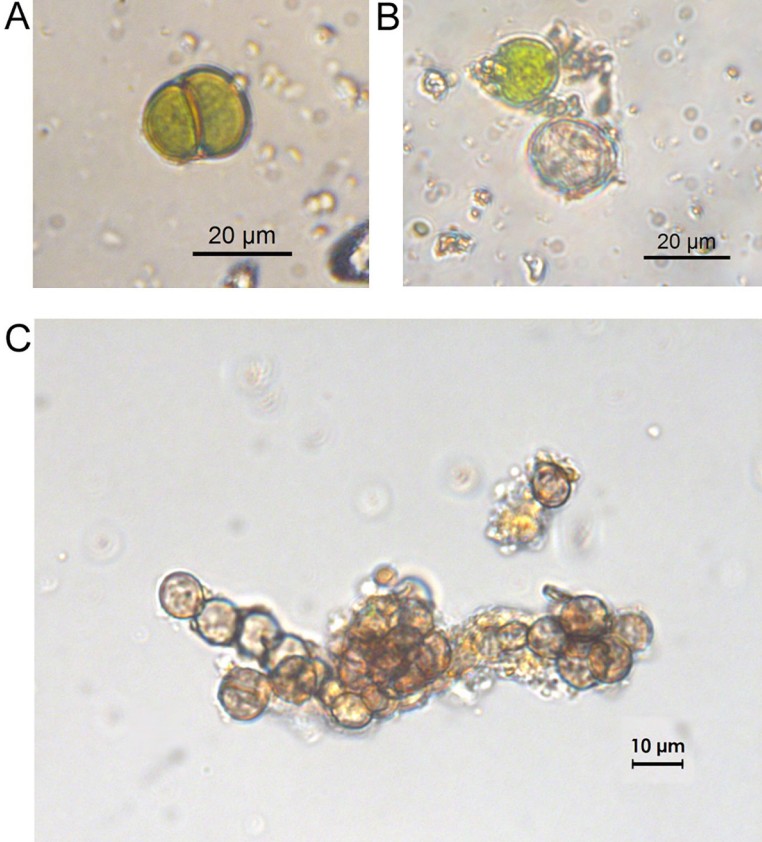

**Fig 2. Optical microscope images of the microorganisms observed in B samples.** (A) *Chroococcus litophilus*
(Cyanobacteria); (B) coccoid green alga (Chlorophyta); (C) meristematic fungus.

remaining 8 samples. From each amplicon 16S rRNA gene sequence library, we obtained
9,736 ± 3,320 and 5,280 ± 2,036 reads, for B and U respectively (after filtering low-quality
reads and chimeras), corresponding to 559 OTUs. The distribution of microbial communities
was evaluated based on beta diversity (Fig 3A), which reflects differences between bacterial
communities, and the results showed that the samples clustered in two well-defined groups.
Alpha diversity, based on the number of observed OTUs and on Chao1/ACE/Shannon/Simp-
son indices, is a parameter that indicates the richness and the biodiversity of the microbial
community in each sample. Black patina samples had a high alpha diversity index value, indi-
cating great species richness, while the uncolonized controls presented a lower bacterial diver-
sity (Fig 3B). Both ecological parameters thus highlighted two significantly different habitats.
Hierarchical Cluster Analysis (HCA) of bacterial phylum, class, family, order, or species was
conducted and confirmed a significant separation among cohorts U and B (P < 0.001). The
results obtained for black patina showed the presence of taxa mainly belonging to the phyla
Bacteroidetes, Cyanobacteria, Proteobacteria, Acidobacteria, and Actinobacteria, with a pre-
dominance of the first three phyla, whereas Firmicutes and Bacteroidetes were enriched in U
samples. In particular, the phylum Cyanobacteria was the most abundant division of the black
patina, comprising approximately 35% of the total reads in all the libraries. It is noteworthy
that the uncolonized controls did not contain any member of Cyanobacteria and the most
abundant phylum (approximately 45%) was Firmicutes, which was absent in B. The B commu-
nity was mainly composed of genera belonging to the families Sphingomonadaceae (genus

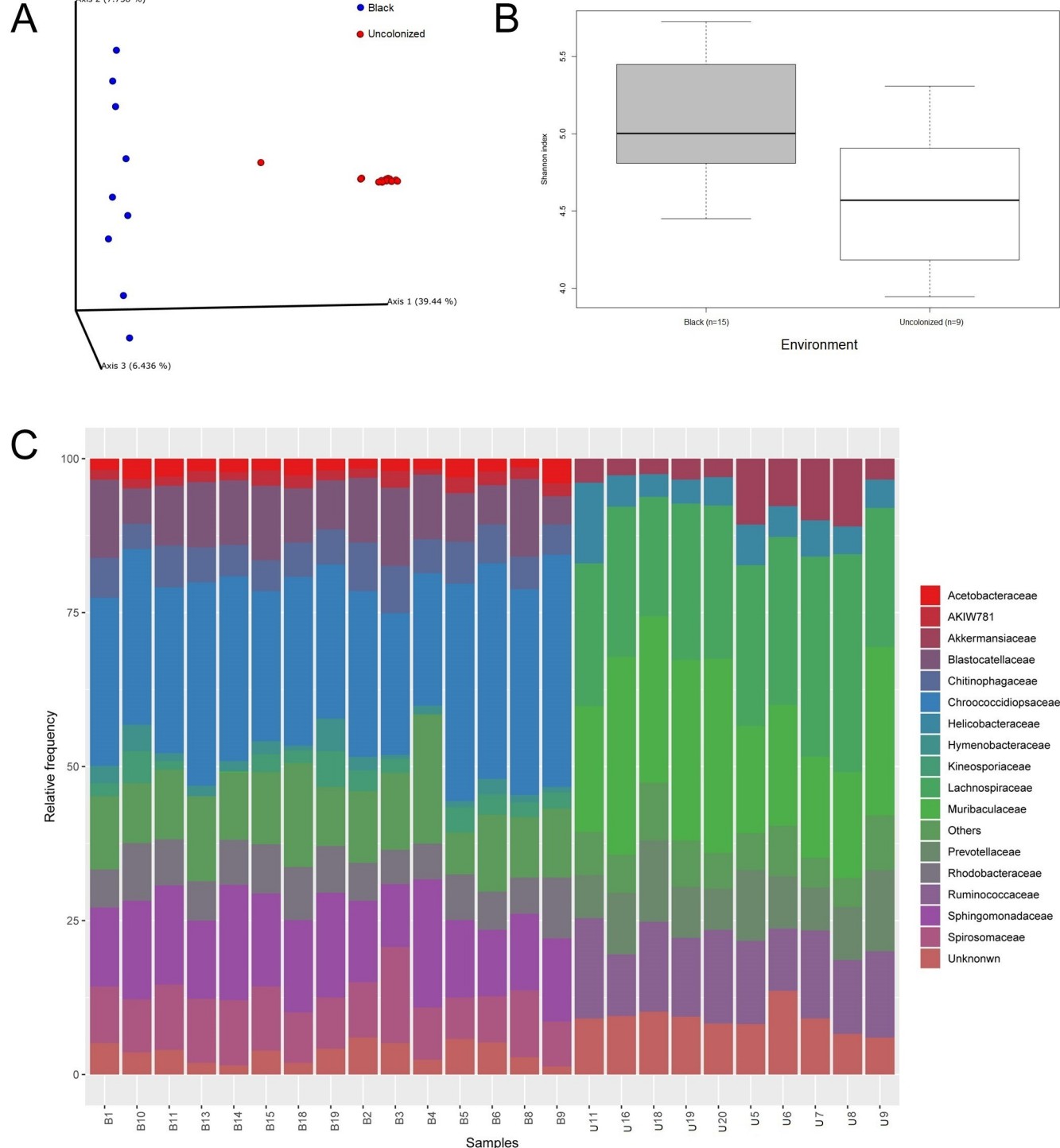

**Fig 3. Statistical analyses of 16S rRNA gene sequencing data.** (A) Beta diversity represented by Principle Coordinate Analysis Emperor plot on a Bray-Curtis distance matrix; (B) Boxplots of Alpha-diversity Index (calculated as Shannon index) in both black patina and uncolonized region samples; (C) Stacked bar charts of the taxonomic profile at family level, low abundance taxa are lumped together in the "Others" category.

*Sphingomonas*), Chroococcidiopsaceae (g. *Chroococcidiopsis* and *Aliterella*), Spirosomaceae (g. *Spirosoma*), Rhodobacteraceae (g. *Rubellimicrobium*), Blastocatellaceae (g. *Blastocatella*),

Chitinophagaceae (g. *Flavisolibacter*), Kineosporiaceae (g. *Quadrisphaera*), and Acetobacteraceae (g. *Craurococcus*) (Fig 3C and S2 Table). The first five families in the preceding list accounted for more than 50% of the identified sequences for ten of the studied samples. The genus *Chroococcus*, observed through optical microscopy in patina samples, was not present in the 16S rRNA gene sequencing results. In order to investigate this absence, we performed an additional BLAST analysis on the unassigned OTUs (S3 Table). These sequences found BLAST hits within the genera *Kryptousia* (10 OTUs), *Nostoc* (7), *Chamaesiphon* (5), *Kastovskya* (3), *Chlorogloeopsis* and *Aliterella* (2), *Vampirovibrio*, *Sinosporangium*, *Hassallia*, *Fischerella*, *Cylindrospermum*, *Chroococcidiopsis*, *Calochaete*, *Brasilonema*, and *Anabaena* (1). Constraining the BLAST search on the genus *Chroococcus* did not give any significant result.

Overall, 16S rRNA gene sequencing confirmed the abundant presence bacteria in the black patina, already observed by optical microscopy, but also demonstrated a high degree of species richness and evenness.

### Fungal community

ITS2-rDNA sub-region amplicon libraries were created from 20 samples of both B and U. Only six U samples were able to enrich for the ITS2 region (S1 Table). The number of sequences for fungi was about 6,707 ± 4,446 reads for B and about 217 ± 389 reads for U samples. Beta diversity, similarly to the bacterial dataset, showed two distinct communities in which the black patina exhibited the greatest biodiversity (Fig 4A).

Sequence analysis of B samples identified more than 80 taxa belonging to the Kingdoms Fungi, Plantae, and Chromista. Fungi were present in all analyzed samples. Excluding the unclassified sequences, the most represented phylum was Ascomycota, with relative frequencies always higher than 40%. Basidiomycota sequences were < 10% for nineteen of the twenty analyzed samples and approximately 17% for the remaining sample. The most represented classes were Dothideomycetes, Eurotiomycetes, Lecarnomycetes, Tremellomycetes, Cystobasidiomycetes, Sordariomycetes, and Agaricomycetes, with a marked predominance of the first two. The most represented orders were Dothideales, Pleosporales, Capnodiales (class Dothideomycetes), Chaetothyriales, Verrucariales (class Eurotiomycetes), Teloschistales (class Lecarnomycetes), Filobasidiales (class Tremellomycetes), Helotiales (class Leotiomycetes), and Lecarnorales (class Lecarnomycetes). The predominant genera were *Coniosporium, Aureobasidium, Caloplaca, Filobasidium, Setophaeosphaeria, and Alternaria* (Fig 4B and S4 Table).

Organisms from the kingdom Plantae were observed only in two of the analyzed samples, with the genus *Trebouxia* (phylum Chlorophyta, class Trebouxiophyceae, order Trebouxiales, family Trebouxiaceae) representing 1% of the analyzed sequences (Fig 4B and S4 Table).

Organisms from the kingdom Chromista were present in two of the B samples. It was not possible to characterize these sequences at a lower level.

All the sequences identified for U samples belonged to the kingdom Fungi, phyla Ascomycota and Basidiomycota. Not considering the unidentified sequences, the only classes represented were Eurotiomycetes, Sordariomycetes, and Malasseziomycetes within the orders Chaetothyriales, Xylariales, and Malasseziales, respectively. Only the genera *Coniosporium* and *Malassezia* were identified (Fig 4B and S4 Table), while for the sequences belonging to the order Xylariales it was only possible to determine the family Diatrypaceae.

### Shotgun sequencing

Amplicon sequencing allowed for the classification of taxa present in the black patina, but in order to further characterize the genetic features of the microbial communities, we performed the shotgun sequencing. Only 13 black patina and 3 control samples generated reads (S1

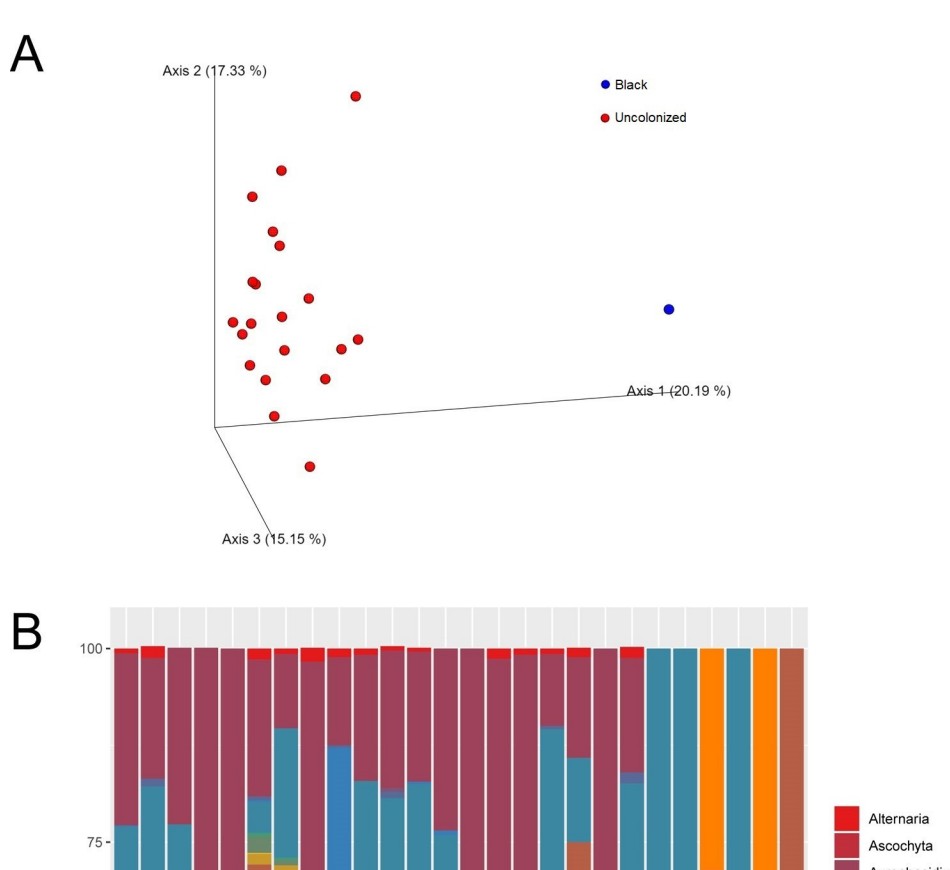

**Fig 4. Statistics about ITS2 data.** (A) Beta diversity represented by a Principle Coordinate Analysis Emperor plot on a Bray-Curtis distance matrix; (B) Stacked bar charts of the taxonomic profile at species level, low abundance taxa are lumped together in the "Others" category.

**Table 1. MAGs genome assembly statistics including completeness and redundancy scores.**

| ID | Putative Classification | Genome size | Number of contigs | N50 | GC % | Completeness | Redundancy |
|---|---|---|---|---|---|---|---|
| Bin_110 | *Hymenobacter sp.* | 5.25 Mb | 971 | 7.572 | 62,71 | 97,12 | 6,47 |
| Bin_113_1 | Cyanobacteria | 5.44 Mb | 897 | 8.401 | 42,11 | 93,53 | 7,91 |
| Bin_95 | *Hymenobacter sp.* | 5.36 Mb | 1.289 | 4.893 | 56,06 | 87,77 | 5,76 |
| Bin_97_1 | Chloroflexi | 5.44 Mb | 1.097 | 6.418 | 58,76 | 87,05 | 7,19 |
| Bin_124 | Chitinophagaceae | 4.24 Mb | 766 | 7.358 | 48,39 | 87,05 | 2,88 |
| Bin_70_1 | Actinobacteria | 2.90 Mb | 364 | 9.563 | 70,07 | 87,05 | 5,76 |
| Bin_126_1 | Acidobacteria | 2.71 Mb | 292 | 12.334 | 63,43 | 86,33 | 7,19 |

Table). Binning with the CONCOCT algorithm resulted in 202 bins, accounting for >702 Mbp, however, after a manual check, the number of genomes dropped to seven (Table 1); two were classified as *Hymenobacter* sp., whereas the remaining five could only be classified at a taxonomic level above genus level (ANI below 85% with the known closest relative). Ten bacterial species (*Friedmanniella luteola*, *Gemmatirosa kalamazoonesis*, *Geodermatophilus obscurus*, *Hymenobacter* sp., *Modestobacter marinus*, *Spirosoma montaniterrae*, and *Spirosoma rigui*) displayed a significantly greater abundance in black patina samples compared to control samples (LDA score above 3.5) (Fig 5). Species with a significantly higher representation in U samples were *Cutibacterium acnes*, *Massilia oculi*, and *Pseudomonas aeruginosa*, along with sequences classified as *Homo sapiens*.

Functional analysis revealed that most of the proteins encoded in the MAGs genomes were classified as unknown functions (COG category "S", excluded from Fig 6 for readability); on the other hand, proteins connected to cell-wall/membrane/envelop biogenesis (COG category "M"), as well as proteins involved in replication and repair machinery (L) and in amino-acid transport and metabolism were highly represented (Fig 6), whereas proteins involved in RNA processing and modification (A), cell motility (N), and cell cycle control and mitosis (D) were

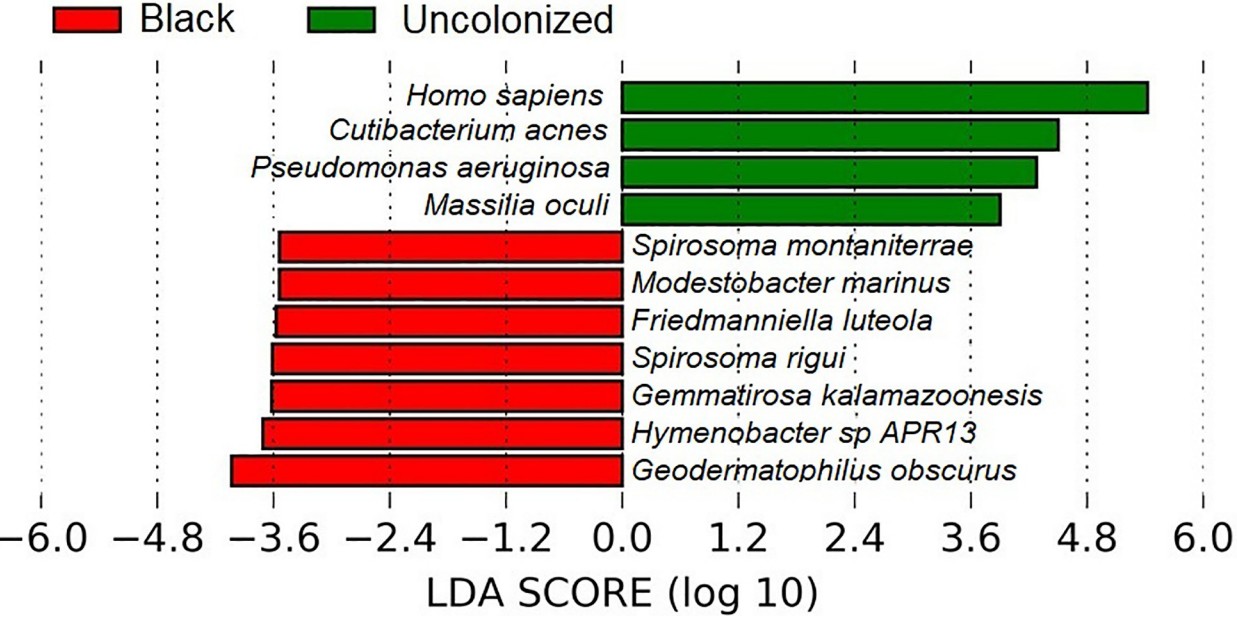

**Fig 5. Species found to be differentially abundant using LEfSe analysis on shotgun metagenomics data analyzed using kraken.**

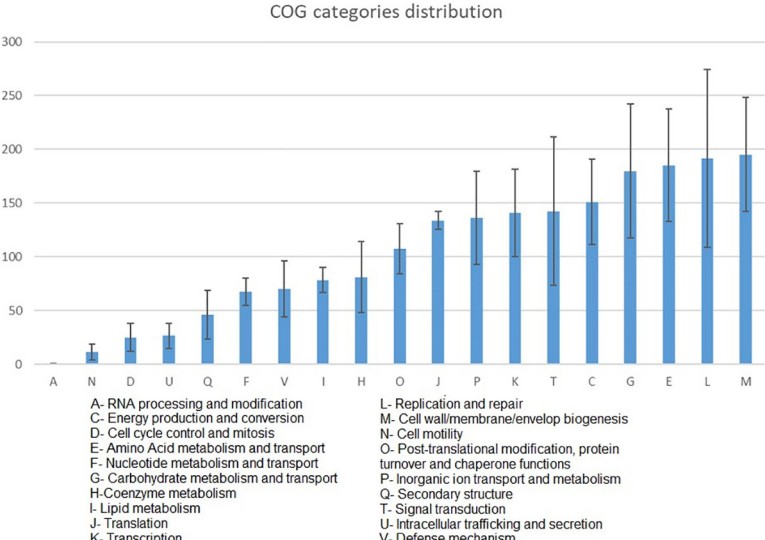

**Fig 6. Bar charts summarizing the number of proteins annotated (y-axis) in a specific COG category (on the x) in the seven MAGs.**

the least represented. No significant differences could be calculated between the two environments because the genomes were all derived from black patina samples.

## Discussion

Next-Generation Sequencing allowed for the characterization of the composition of black patina on the travertine Tiber embankments. In particular, it was possible to show that the black patina samples displayed higher diversity than controls, which was confirmed by 16S rRNA gene and ITS2 amplicon sequencing.

The bacterial communities of B samples were dominated by cyanobacteria, which accounted for 35% of the total reads in all libraries. Ogawa et al., (2017) [46] reported that the proportion of cyanobacteria in sequenced reads from decorative siliceous stone was unexpectedly low and proposed the use of specific primers for detecting this phylum. We did not observe this failure at phylum level and the present results indicate that 16S rRNA amplicon sequencing can successfully be used to analyze microbial communities inhabiting stone monuments.

Among the bacterial families identified in B samples, only Chroococcidiopsidaceae and Nostocaceae are mentioned in the literature with genera (e.g. *Gloeocapsa*, *Chlorogloea*, *Myxosarcina*, *Rivularia*, *Tolypothrix*, and *Nostoc*) that are well known members of microbial patinas on rocks in subaerial environments [4,16,47]. In particular, the identified genera *Chroococcidiopsis* and *Scytonema*, belonging to these families, have often been reported in studies of the colonization of rock surfaces, frescos, plasters, and limestone monuments [19,47]. Cyanobacteria are an important component of subaerial biofilms and usually dominate communities where water seepages are present [8]. Their survival on rock surfaces is guaranteed due to their ability to withstand desiccation and the presence of pigments and other UV-protectants in the cytoplasm or EPS [12,48–50]. In the most extreme environments, some cyanobacteria have the ability to avoid environmental stresses by living inside the rock (endolithic habitus); some species belonging to the genus *Chroococcidiopsis* are considered the most frequent and widespread crypto-endolithic organisms [51].

In order to understand the biodegradative role of the black patina, it must be recognized that several studies have highlighted the importance of cyanobacteria in the weathering of rocks [52–55]. In particular, Pentecost (1992) [56] reported that black patinas containing cyanobacteria belonging to the genera *Gleocapsa* and *Scytonema* may play a key role in this process, achieving a surface weathering rate of up to 3 mm/100 y, probably because the dehydration and rehydration of the sheaths contribute to the loosening of the rock.

Bacteria of the family Sphingomonadaceae are commonly isolated from soil, freshwater, and marine habitats, and from plant phyllospheres or rhizospheres, with few species reported as human or plant pathogens [57]. Species of the genus *Sphingomonas*, identified in all the B samples, are involved in polycyclic aromatic hydrocarbon degradation [57,58], and their presence in the black patina may therefore be linked to the presence of substances of anthropogenic origin (e.g. pollutants from motor vehicles). Blastocatellaceae comprises mesophilic and thermotolerant bacteria that are slow-growing K-strategists that prefer oligotrophic growth conditions and are able to survive drought and nutrient limitation; species belonging to this family are common in arid soils and soil crusts [59]. Spirosomaceae includes gram-negative, ring-forming, aerobic, nonmotile bacteria, mainly isolated from aquatic (freshwater lakes, marshes, and marine water) and soil environments; members of this family produce characteristic non-diffusible pigmentation [60]. Rhodobacteraceae comprises mainly aquatic bacteria, frequently isolated from marine environments, including aerobic photo- and chemo-heterotrophs or anaerobic photoautotrophic purple non-sulfur bacteria [61]. Bacteria belonging to Chitinophagaceae are gram-negative, often forming yellow colonies [62]; in particular, species of the genus *Flavisolibacter* have been isolated mainly from soil and water [63–65]. Finally, Kineosporiaceae comprises a group of diverse aerobic mesophilic actinobacteria, isolated mainly from soils and plant materials [66]; the genus *Quadrisphaera* includes microorganisms isolated from a batch-fed activated sludge reactor [67].

Considering that most of the microorganisms belonging to the identified families originate from water or soil environments, their presence in the black patina is as a result of contamination due to the proximity of the Tiber and its shores cannot be excluded. However, the high relative abundance of some of the identified genera (reaching almost 20% in the case of *Sphingomonas*) led us to hypothesize that the presence of these microorganisms in the patina was not due to contamination but that they actually formed part of the patina. The fact that these microorganisms have never before been reported in black patinas on stone artifacts is most likely due to the techniques routinely used in the field of cultural heritage, which are often not appropriate for their detection or identification [68,69].

Some of the bacterial species identified in B samples through shotgun sequencing deserve special attention. *Spirosoma montaniterrae* are gram-negative, yellow-pigmented, long rod-shaped bacteria, highly resistant to UV-C and gamma-radiation. They have been isolated from a mountain soil sample collected at Mt. Deogyusan (Jeonbuk province, South Korea) [70]. *Hymenobacter* is a gram-negative, non-motile bacterial genus belonging to the family Cytophagaceae (order Sphingobacteriales). Several species belonging to this genus have been isolated from extreme environments (e.g. Antarctic soil, sandstone surface, permafrost, uranium mine waste waters, etc.) and exhibit high radiation resistance [71–74]. Both the 16S rRNA gene sequence and the shotgun sequencing data sets revealed *Hymenobacter* as one of the differentially abundant taxa between control and black patina samples (more abundant in B). It was also possible to assemble the whole genome for this taxon; this implies that it had both a specific tetranucleotide signature and pattern of coverage across samples. Finally, *Geodermatophilus obscurus* and *Modestobacter marinus*, belonging to the family Geodermatophilaceae (order Geodermatophiliales), are described in literature as stone-dwelling actinobacteria resistant to environmental hazards, involved in stone biodeterioration due to their euendolithic

habitus [75,76]. Geodermatophilaceae are usually isolated from desert soil or rock varnish in deserts [77–81], *G. obscurus* and species of the genus *Modestobacter* have also been isolated from black, orange and grey patinas of stone monuments in the Mediterranean basin [82]. Proteogenomic analysis of these two species carried out by Sghaier et al. (2016) [83] investigated their adaptation strategies to the stone-surface environment. The work highlighted the presence of several genes, involved in the biosynthesis of carotenoids (absorbing almost uninterruptedly from 200 nm to 750 nm), stress relief, reactive oxygen species detoxification, and DNA protection and repair, in addition to operons encoding genes for photosynthesis reactions, and highly expressed biomarkers encoding proteins implicated in the development of biofilms. It is interesting to note that the functional analysis carried out during our study underlined the presence of proteins connected to strategies of adaptation to harsh and stressful environments. In particular, cell-wall, membrane, and envelope biogenesis proteins (COG category "M"), as well as proteins involved in replication and repair machinery (L), are indicative of bacterial response to environmental stresses like UV radiation and desiccation [84–86].

Fungal species belonging to several of the genera identified in the B communities (e.g. *Aureobasidium pullulans*, *Coniosporium apollinis*, *Alternaria alternata*, *Cladosporium*, *Knufia petricola*, and *Vermiconia antartica*) are reported as RIF and MCF, isolated from patinas on monuments or stone surfaces in urban environments [9,87–94]. It is well known that these fungi are able to survive the harsh conditions of stone surface due to several adaptations. For example, melanin pigmentation confers mechanical strength to hyphae enabling the fungi to grow into crevices, and, together with other pigments (carotenoids and mycosporines), protects the cells from UV radiation [8,95]. Furthermore, the microcolonial and yeast-like growth makes the cells thermodynamically efficient, and able to withstand heat and desiccation [8].

The genera *Aureobasidium*, *Alternaria*, and *Cladosporium* are typical of fungal communities in moderate or humid climates, while others like *Coniosporium*, are characteristic of arid and semi-arid environments [88]. This could identify the Tiber's embankments as an intermediate habitat. In order to define the biodegradative role of the black patina, it must be underlined that MCF are able to penetrate rock substrates mechanically, producing lesions known as pitting [93,95–97]. This action, linked to the forces exerted by growing hyphae, is made possible by the rigidity of the cell wall and by the turgor of the cells. A study recently carried out by De Leo and colleagues (2019) [6] on a marble statue of the Quirinale Palace's Gardens (Rome) showed that MCF (*Coniosporium apollinis* and *Knufia* sp.) penetrated the marble to a depth of more than 100 μm, producing tunnels of up to 11 μm in diameter, disposed along the marble crystals' edges or deeply penetrating inside them. It is interesting to note that the abundance of MCF genera in black patina from this study was much lower than reported in other studies [e.g. 9]. In order to reject the hypothesis that these conflicting results were linked to the absence of MCF sequences in the QIIME2 database, the database was mined for sequences from genera most frequently reported in the studies on RIF. Apart from a few exceptions, all considered genera were present in the database with one or more species. Therefore, the scarcity of MCF sequences in the Tiber embankments' black patina can be attributed, with some certainty, to the actual composition of the community. The discrepancy between MCF data from this study and literature data might be linked to the shading conditions created by plants that influence moisture content and surface temperatures, creating an environment not suitable for the growth of most of the MCF species adapted to more extreme environments. Furthermore, it is known that the presence of vegetation in the vicinity of the stone surface colonized by black patina microorganisms, influences the composition of the community [9,21]. The occurrence in the B samples of genera containing wood-associated, saprobic, and plant pathogen species (e.g. *Erysiphe*, *Filobasidium*, *Setophaeosphaeria*, *Sclerostagonospora*, *Coprinopsis*, and *Macrodiplodiopsis*) [98–104] can therefore be easily explained as

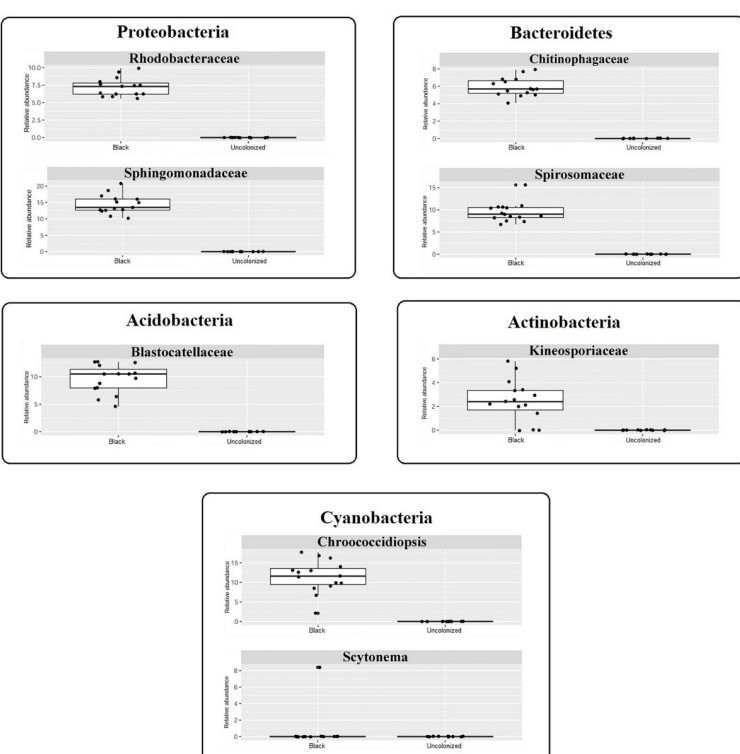

**Fig 7. Boxplots showing the differential abundance of specific bacterial taxa.** The graphics are divided according to the phyla they belong.

contamination, linked to the presence of the riparian vegetation, mainly composed of trees belonging to the genus *Platanus* L., 1753. Plant-related genera, species of which are reported in literature as phylloplane yeasts (e.g. *Filobasidium*, *Buckleyzyma*, *Papiliotrema*, *Vishniacozyma*, and *Dioszegia*) [105–107], deserve a special mention. Phylloplane microorganisms are adapted to live in the harsh conditions of plant leaf surfaces, conditions mainly characterized by poor and fluctuating nutrient availability and relatively long periods of desiccation and high solar radiation due to the production of protective pigments such as melanin, mycosporines, ubiquinone, and carotenoids [107,108]. These adaptations are very similar to those usually reported for MCF adaptation on rock surfaces. Taking these features into consideration and the fact that the relative abundance of some of these genera in Tiber embankment black patina was greater than 6%, it is hypothesized that phylloplane yeasts could form an active part of black patina, and that their presence was not simply due to riparian vegetation contamination. Corroborating this hypothesis is the fact that *Aureobasidium pullulans*, normally reported as MCF typical of black patinas, has also been reported as a phylloplane yeast [107].

ITS2-rDNA sub-region amplicon analysis revealed the presence of the lichen genera *Caloplaca* and the Chlorophycean *Trebouxia*. *Caloplaca* species are colonizers of base-rich siliceous stones, limestone, concrete, sandstone, and mortar [109–112], yet to our knowledge there are no reports on the presence of this genus in black patinas. *Trebouxia* is a unicellular green algae; *T. arboricola* has often been reported as an epiphytic or epilithic colonizer of stone monuments in shaded areas characterized by low substrate humidity or prone to desiccation [113–115].

Figs 7 and 8 clearly show that the composition of the microbiota of U samples is very different to that of black patina. The bacterial and fungal taxa most abundant in B samples are

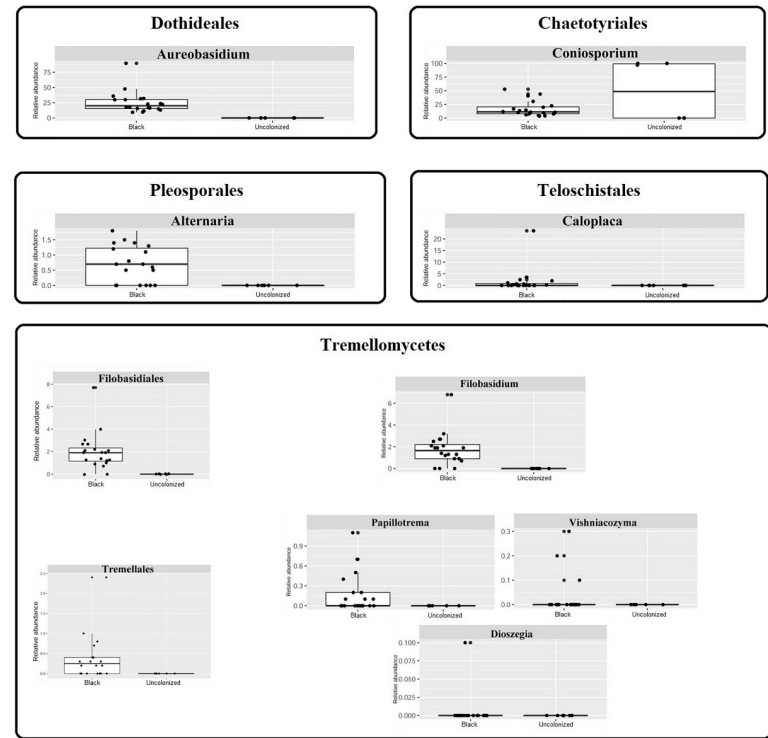

**Fig 8. Boxplots showing the differential abundance of specific fungal taxa.** The graphics are divided according to the order they belong.

almost completely absent in the control areas. It is worth noting that in U almost all the families identified through 16S rRNA gene sequencing (e.g. Lachnospiraeae, Muribaculaceae, Prevotellaceae, Ruminococcaceae, Helicobacteraceae, and Akkemansiaceae) are typically found in the mammalian gut environment or in human feces [116–121]. Furthermore, of the three species identified through shotgun sequencing, *Pseudomonas aeruginosa* and *Cutibacterium acnes* are environmental and commensal human skin bacteria (causing opportunistic infections), respectively [122,123], while *Massilia oculi* has reportedly been isolated from a clinical specimen from a patient suffering from endophthalmitis [124]. Regarding ITS2-rDNA sub-region amplicon analysis data, *Coniosporium* sp. are considered a RIF and the genus *Malassezia* comprises species related to opportunistic infections, commonly find on the skin of many animals, including humans, [125]. Furthermore, Diatrypaceae comprises species predominantly saprotrophic on the decaying wood of angiosperms worldwide, which are sometimes associated with plant diseases [126]. Considering the low number of reads and the few identified taxa, we cannot say that U samples contained actual bacterial or fungal communities. The presence of bacteria in U samples is due to obvious contamination from the Tiber's water or from humans frequenting the river's bank, while the presence of fungal sequences is the result of environmental contamination.

When comparing the results obtained with optical microscope observations and NGS, the difference in sensitivity between optical microscopy, that allowed the observation and identification of only limited taxa, and NGS, that identified > 100 different taxa in the black patina, is very obvious. Molecular analyses confirmed the predominance of bacteria and cyanobacteria in the black patina and of bacteria in the U samples, as observed through optical microscopy. The 16S rRNA and ITS2-rDNA sub-region sequencing allowed the identification (to genus

level) of the microorganisms observed using optical microscopy (for example *Chroococcidiopsis* and *Scytonema* for the coccoid and filamentous cyanobacteria, respectively, *Coniosporium*, *Aureobasidium*, and *Knufia* for the MCF, and *Trebouxia* for the coccoid green algae), and highlighted the presence of lichens not observed with the naked eye or microscope. Currently, the main limitation of NGS in studying black patinas and subaerial biofilms in general, is the absence of several OTUs in the reference database. This is confirmed by the presence of the unidentified sequences in the obtained results.

The absence of 16S rRNA gene amplicons of *Chroococcus* species, which contradicts the microscope observations, could be due to the lack of corresponding sequences of this species in the databases. While the lack of *Chroococcus* in the assembly with NCBI of unassigned reads may be related to specific difficulties in DNA extraction.

It is important to note that the complete overview of the black patina composition obtained with NGS is unique compared to the results reported in literature. In fact, in the studies carried out on black patinas, identification of microorganisms was achieved by optical microscopy or DNA sequencing and usually focused on cyanobacteria, algae, fungi, and lichens, completely neglecting bacteria. As also shown in the present work, microscope analyses, in most cases, do not yield good results. Phototrophic microorganisms have a high morphological plasticity that sometimes makes identification impossible, while RIF, and in particular MCF, have no taxonomical features useful for distinguishing different genera [e.g. 4,6]. To overcome these limitations the most recent studies have used molecular techniques. However, in these studies DNA for sequencing was extracted from cultured microorganisms [e.g. 9,88,89] and it is well known that culture methods are highly biased by the living condition of the analyzed microorganisms, which are adapted to live in harsh environments that are difficult to reproduce in the laboratory. For this reason, data generated using such methods are only representative of a selection of species that are more easily cultivable and not of the whole community.

Knowledge of the composition of the black patina is useful for conservation purposes. NGS results highlighted the presence of microorganisms involved in stone weathering (cyanobacteria) or with euendolithic habitus (MCF). This emphasizes that the role of black patina in the biodeterioration of the stone artifacts is not just linked to an aesthetical alteration and should not be neglected. Periodic restoration interventions should probably be carried out to prevent microorganisms from degrading the artifacts. Furthermore, considering the ability of NGS to detect the presence of biodeteriogens, even when low amounts of DNA are present, this molecular technique could allow the monitoring of stone colonization over time, even after restoration interventions.

## Conclusions

The high-throughput sequencing study of the black patina of the Tiber's embankments highlighted the rich diversity of bacterial and fungal communities. The method allowed for the collection of extensive information on the total (culture-independent) microbial community comprising microorganisms which have thus far been underestimated or neglected in analyses routinely carried out in studies of stone artifacts. Furthermore, the sequencing results allowed for the identification of genera that could not be identified through optical microscopy due to a lack of defining morphological features.

The sequencing results gave a clear idea of the microbial composition of the Tiber's embankments' black patina, highlighting the presence of genera described as endolithic biodeteriogens, or weathering agents. The obtained results emphasize how in-depth knowledge of the patina composition allows an understanding of its important biodeteriogenic role, which is often underestimated. Furthermore, the NGS analyses highlighting the presence of taxa not

detected in previous studies may serve as a starting point for further investigations on the bio-deteriogenic role of these groups of microorganisms.

## Supporting information

**S1 Fig.** Rarefaction curves showing the Shannon values (A and B) and phylogenetic diversity (C and D) for ITS (A and C) and 16S rRNA (B and D). Values are grouped according to the type of samples (black patina vs. uncolonized controls) and displayed as box plots each with 500 reads.
(TIF)

**S1 Table. Metadata: Distinct sample IDs, specific amount of DNA extraction and corresponding obtained library.** Red color indicates a failed first quality check, whereas an asterisk (\*) shows the samples excluded after rarefaction curve analysis.
(DOCX)

**S2 Table. Average taxonomic abundance (with standard deviation) for each bacterial family in uncolonized controls and black patina samples.**
(DOCX)

**S3 Table. Taxonomic assignment inferred by BLAST on the NCBI 16S ribosomal sequence database of unclassified reads at the level of phylum (unclassified cyanobacteria) and kingdom (unclassified bacteria).**
(DOCX)

**S4 Table. Average taxonomic abundance (with standard deviation) for each fungal genus in uncolonized controls and black patina samples.**
(DOCX)

## Acknowledgments

The authors would like to thank Dr. Barbara Davidde Petriaggi and Dr. Luca Pozzana for their precious support.

## Author Contributions

**Conceptualization:** Federica Antonelli, Francesca Guerrieri.

**Data curation:** Alfonso Esposito, Silvano Piazza.

**Funding acquisition:** Francesca Guerrieri.

**Investigation:** Federica Antonelli, Ludovica Calvo, Francesca Guerrieri.

**Methodology:** Ludovica Calvo, Sandra Ricci, Francesca Guerrieri.

**Project administration:** Francesca Guerrieri.

**Resources:** Federica Antonelli, Sandra Ricci, Silvano Piazza, Francesca Guerrieri.

**Software:** Alfonso Esposito, Silvano Piazza.

**Supervision:** Francesca Guerrieri.

**Validation:** Valerio Licursi, Philippe Tisseyre, Manuela Romagnoli.

**Visualization:** Federica Antonelli, Alfonso Esposito.

**Writing – original draft:** Federica Antonelli, Sandra Ricci, Francesca Guerrieri.

**Writing – review & editing:** Federica Antonelli, Alfonso Esposito, Francesca Guerrieri.

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
