## [Decision Letter · Decision Letter 0]

11 Oct 2019

PONE-D-19-21352

Tiber’s embankments black patina characterization by Next-Generation Sequencing

PLOS ONE

Dear Dr Guerrieri,

Thank you for submitting your manuscript to PLOS ONE. After careful consideration, we feel that it has merit but does not fully meet PLOS ONE’s publication criteria as it currently stands. Therefore, we invite you to submit a revised version of the manuscript that addresses the points raised during the review process.

Although this study is interesting, some of the criteria publication have not been achieved in particular namely criteria 3, 4 and 5 (https://journals.plos.org/plosone/s/criteria-for-publication). Thus, the authors are strongly advised to hire a copyeditor. In particular, sections such as Abstract/Results and Discussion must be improved, as referred by the reviewers.

It is important as indicated by reviewers to give more detail in the material and methods section “DNA extraction, library preparation and sequencing”. In addition to the reviewer’s suggestions I would also advise the authors to indicate the range of the concentration of DNA obtained for the samples. It is also important to know the rarefaction values for Beta diversity.

Discussion must be improved as mentioned by the reviewers. Additionally, I would like to know if we can exclude handling contamination when you discuss the following results: “Considering the low number of reads and the few identified taxa we cannot say that actual bacterial or fungal communites were present in the W samples. The presence of bacteria is due to a conspicuous contamination from the Tiber’s water or from humans frequenting the river’s bank, while the presence of fungal sequences is just linked to the environmental contamination.”?

Because of the negative result mentioned “this allowed, for the first time, to explore the whole structure of a black patina bypassing the culture problem linked to the underestimation of unculturable species and comprising the bacteria that are usually neglected in routinely analyses. The absence in the molecular results of Chroococcus lithophilus, identified through microscope observation, can be linked to the lack of 16S sequences of this species in the databases. Indeed, currently the main limit to the application of NGS to the study of black patinas and, more in general, of subaerial biofilms is linked to the absence in the databases of several environmental microorganisms’ sequences (like cyanobacteria and RIF). This is confirmed by the presence of several unidentified sequences in the obtained results.”. Have you tried to blast/map the unidentified sequences with the 16S sequence of Chroococcus lithophilus from for example ncbi or EzBioCloud?

Minor issues:

Standardize in the text “(g, Chroo…” or “(g. R..)”, but not both;

Standardize in the text “(species A. pullulans)” or “(A. alternata)”, but not both;

We would appreciate receiving your revised manuscript by Nov 25 2019 11:59PM. To enhance the reproducibility of your results, we recommend that if applicable you deposit your laboratory protocols in protocols.io, where a protocol can be assigned its own identifier (DOI) such that it can be cited independently in the future. For instructions see: http://journals.plos.org/plosone/s/submission-guidelines#loc-laboratory-protocols

We look forward to receiving your revised manuscript.

Kind regards,

Ana R. Lopes, PhD

Academic Editor

PLOS ONE

1. In your Methods section, please provide additional information regarding the permits you obtained for the work. Please ensure you have included the full name of the authority that approved the sampling site access and, if no permits were required, a brief statement explaining why.

2. We note that you are reporting an analysis of a microarray, next-generation sequencing, or deep sequencing data set. PLOS requires that authors comply with field-specific standards for preparation, recording, and deposition of data in repositories appropriate to their field. Please upload these data to a stable, public repository (such as ArrayExpress, Gene Expression Omnibus (GEO), DNA Data Bank of Japan (DDBJ), NCBI GenBank, NCBI Sequence Read Archive, or EMBL Nucleotide Sequence Database (ENA)). In your revised cover letter, please provide the relevant accession numbers that may be used to access these data. For a full list of recommended repositories, see http://journals.plos.org/plosone/s/data-availability#loc-omics or http://journals.plos.org/plosone/s/data-availability#loc-sequencing.

Reviewers' comments:

Reviewer's Responses to Questions

**Comments to the Author**

1. Is the manuscript technically sound, and do the data support the conclusions?

Reviewer #1: Partly

Reviewer #2: Yes

Reviewer #3: Yes

2. Has the statistical analysis been performed appropriately and rigorously? 

Reviewer #1: Yes

Reviewer #2: Yes

Reviewer #3: Yes

3. Have the authors made all data underlying the findings in their manuscript fully available?

Reviewer #1: Yes

Reviewer #2: Yes

Reviewer #3: Yes

4. Is the manuscript presented in an intelligible fashion and written in standard English?

Reviewer #1: Yes

Reviewer #2: Yes

Reviewer #3: No

5. Review Comments to the Author

Reviewer #1: Black patina are common features in monuments and walls around the world. They often associated with substrata where microenvironmental conditions promote the retention of water. They can be broadly classified depending on the dominant process leading to their formation. Earlier studies highlighted the chemical, pollutant-related origin of black crusts, often associated with Sulphur-laden atmosphere and the formation of gypsum layers that harden on the stone upper layers. On the other hand, on the Tropics, the occurrence of biologically-dominated black crust have been recently reported to be predominantly linked to microbial colonization by pigmented organisms synthesizing compounds such as scytonemins and mycosporine-like substances. It appears that the authors have dealt with in this study with latter type of black crusts; they should highlight this in the title. Secondly, I find that the abstract fails to display the main findings. NGS studies about subaerial communities are comparatively less studied than other terrestrial habitats, the authors should take advantage of this gap of knowledge and exploit more their data. Third, the introduction could certainly be improved by adding relevant references that highlight the biological composition of black crust in stone surfaces. I feel that the authors could improve their study by providing more details about the nature on substratum (mineralogy of stone, from bibliographic data), the prevailing microenvironment (from nearly meteorological stations) and orientation of surfaces. I could notice that samples were taken from either Black or White areas (replicates). A this point I am not sure the white-derived samples are originated from a “white platina” or non-colonized (at least by visual inspection) surfaces. A patina normally refers to surface alteration by a number of processes that result on modification of the upper layer, sometimes pure staining (aesthetic) but also chemical transformation of the upper layer. Please clarify this point. Also, please state if they were taken randomly. In addition, indicate how you managed with graffiti that is obvious on the image. Adding arrows to the sampled areas would have the reader to clearly identity the origin of samples. The apparent lack of correspondence of certain organisms not identified by NGS but seen by microscopy could be explained by the non-efficient extraction of nucleic acids, we have seen this in the past with thick-sheathed cyanobacteria.

Overall, I find that the results need be better contrasted with published studies based on both non-culture dependent and culture-dependent studies from epilithic habitats and highlight the main findings. Please also state and provide relevant references as to how this type of NGS-based study can provide relevant information regarding conservation issues. In addition, the conclusion section needs to be enhanced to fully be supported by the results.

Reviewer #2: The authors present an interesting study by analyzing the microbiota of a black patina often found over travertine embankments of Tiber river in Rome. For this reason, Next-Generation sequencing techniques, through Illumina platform, were applied in order to identify and characterize different communities of bacteria, fungi and algae, as a mean to understand the possible effects of these colonial organisms to the studied material. The study is well organized, presents relevant data, especially the statistics results and graphics, and the manuscript is well written. I have only minor comments that can be found in comments below:

Introduction

Line 48 – replace “works of art” for “artworks”

In this section a final paragraph with the study objectives is missing. Please add the objectives of the work to complete well the Introduction section.

Materials and Methods

Line 103 – replace the number “1” for number “2”. In this manuscript “Materials and Methods” are the section number 2.

Results

In sub-sections “3.1 Bacterial Community” and “3.2 Fungal community” please add the percentages of abundance of the described taxa. This is relevant data in such NGS study and is missing on this section.

Discussion

I advise the authors to add an introductory paragraph to this section, instead of starting immediately with the results discussion. It would be good to start with some statement (3-4 lines) regarding the importance of the used methodology to characterize the microbiota communities of the black patina present in such important Cultural Heritage structure, which was actually the aim of this study.

Reviewer #3: I believe that the manuscript by Antonelli et al. “Tiber’s embankments black patina characterization by Next-Generation Sequencing” (ref: PONE-D-19-21352) is a very interesting study concerning the complete metagenomic analysis of black patinas in an important stone monument. In my opinion, the topic is relevant and deserves to be highlighted. I also would like to pinpoint that the application of Shotgun metagenomics is currently rather scarce in the field, thus turning the article highly innovative. I recommend the acceptance of the article after the authors conduct major revisions in the manuscript, and some crucial points are addressed. I´m providing some comments to be taken into consideration by the authors:

(1) Comment: The article should be proofread by an English native speaker.

(2) Comment: The term 18S ITS should be replaced for ITS2-rDNA sub-region, since from my understanding the 18S region (SSU) was not considered during the course of this study.

(3) Introduction section, Lines 49-60 and Lines 75-76.

Comment: Please consider rephrasing these sentences. They are too long and their structure could be improved.

(4) Line 71 and Line 284.

Comment: In line 71. the reference for Pentecost (1992), in this case [45], is missing. In line 284, the reference for Albertano (2012), in this case [18], is also missing. I advise the authors to double check their references along the manuscript and in the references.

(5) Introduction section, Line 64-84.

Comment: This part of the introduction section is only focussed in Phototrophic microorganisms and Fungi. I believe that the role (if any) and presence of bacteria in black patinas (if previously studied), should also be highlighted in this part.

(6) Introduction section, Line 93-95.

Comment: The aims of the study should be clearer.

(7) Introduction section, Line 95-99.

Comment: I believe this part should be moved to the Materials and methods- Sample collection and description sub-section.

(8) Materials and methods section, Line 103. Discussion section, Line 274.

Comment: In line 103, this section should be: 2. Materials and methods. In the current form is displayed as 1. Materials and methods. In line 274, this section should be: 4. Discussion. In the current form lacks numbering.

(9) Materials and methods section; 2.1 Sample collection and description.

Comment: I believe that the manuscript could benefit from the addition of a table displaying the distinct samples IDs and further metadata. The table could also display which samples were able to be studied through the distinct metagenomic methodologies applied.

(10) Materials and methods section; 2.1 Sample collection and description.

Comment: For the microscopical analysis, were the samples randomly selected? What were the criteria for the selection of these samples? Which samples (ID) were studied?

(11) Materials and methods section; 2.2 DNA extraction, library preparation and sequencing.

Comment: Please provide further details regarding the DNA extraction, library preparation and sequencing.

(12) Results section; 3.2 Bacterial community, Lines 204-208.

Comment: This part should be moved to the discussion section.

(13) Results (e.g. Lines 227-230), Discussion (e.g. Lines 341-383) and Figures 4 and 8.

Comment: These parts highlight my main concern with the manuscript. In general, the application of the Illumina MiSeq methodology targeting the ITS2 rDNA sub-region does not allow a proper and accurate taxonomic annotation to the species level. I believe that these parts as well as the figures above mentioned, need to highlight a taxonomic annotation at the genus level, and therefore require to be updated. I don´t feel that the discussion bulk will be affected by this decision. However, I do acknowledge that this change will require several parts of the manuscript to be updated.

(14) Figure 6 legend needs further information, namely the distinct COG categories.

(15) Discussion section.

Comment: Given the quality of the results, some parts of the discussion section could be deeper debated. For example, the authors state that some bacteria taxa can be linked to the presence of substances of anthropogenic origin. Pollution is known to act as an ecological pressure on cyanobacterial communities. Could the presence of these cyanobacterial taxa be linked to extremotolerance profiles allowing them to thrive in these conditions? MCF are also known for their tolerance to various environmental factors, could their presence in the B samples also be explained by their high metabolic capacities?

6. PLOS authors have the option to publish the peer review history of their article (what does this mean?). If published, this will include your full peer review and any attached files.

Reviewer #1: No

Reviewer #2: No

Reviewer #3: No

---

## [Author Response · Author response to Decision Letter 0]

25 Nov 2019

Point-by-point response to reviewer’s comments

Reviewer # 1

We appreciated the challenging quality of the Reviewer 1’s comments that prompted us to

carefully revise the manuscript to clarify experimental details and results in order to increase

its readability and to better support our conclusions.

Specific points

Question 1. Black patina are common features in monuments and walls around the world. They

often associated with substrata where microenvironmental conditions promote the retention of water.

They can be broadly classified depending on the dominant process leading to their formation. Earlier

studies highlighted the chemical, pollutant-related origin of black crusts, often associated with

Sulphur-laden atmosphere and the formation of gypsum layers that harden on the stone upper

layers. On the other hand, on the Tropics, the occurrence of biologically-dominated black crust have

been recently reported to be predominantly linked to microbial colonization by pigmented organisms

synthesizing compounds such as scytonemins and mycosporine-like substances. It appears that the

authors have dealt with in this study with latter type of black crusts; they should highlight this in the

title.

Answer 1. We apologize if the description of black patina, and its differences with black

crusts, were not clearly defined in the introduction. Reviewer 1 raises an important point

and, albeit we feel that the definition “black patina” is already a clear statement on the

biological origin of the alteration, in the new MS we highlighted that black crusts and black

patinas are two distinct deterioration patterns of stone artefacts. Black crusts have a

chemical origin and are defined as “Kind of crust developing generally on areas protected

against direct rainfall or water runoff in urban environment. Black crusts usually adhere firmly

to the substrate. They are composed mainly of particles from the atmosphere, trapped into

a gypsum (CaSO4.2H2O) matrix” (Vergès-Belmin V. Illustrated glossary on stone

deterioration. ICOMOS. 2008). The term patina refers to several kinds of alterations, of

organic and inorganic origin, tightly adhering to the substrate. As reported in the

manuscript’s introduction, since the 1990s the term is also used to define an aesthetic

change of rock surfaces linked to the biological colonization. Black patinas are a well-known

biological alteration of stone surfaces studied since the beginning of 1900 in natural

environments. In the field of the conservation of Cultural Heritage several studies covered

this topic in the last decades.

Question 2. Secondly, I find that the abstract fails to display the main findings. NGS studies about

subaerial communities are comparatively less studied than other terrestrial habitats, the authors

should take advantage of this gap of knowledge and exploit more their data.

Answer 2. We thank the Reviewer 1 for this suggestion. The abstract was modified.

Question 3. Third, the introduction could certainly be improved by adding relevant references that

highlight the biological composition of black crust in stone surfaces.

Answer 3. The biological composition of the patina is now better described in the

introduction and further information were added concerning the bacteria composition.

Question 4. I feel that the authors could improve their study by providing more details about the

nature on substratum (mineralogy of stone, from bibliographic data), the prevailing microenvironment

(from nearly meteorological stations) and orientation of surfaces.

Answer 4. We agreed with the Reviewer 1 and the new materials and methods section

contains the data concerning the composition of travertine and thermo-hygrometric trends

registered in the city.

Question 5. I could notice that samples were taken from either Black or White areas (replicates). A

this point I am not sure the white-derived samples are originated from a “white platina” or noncolonized

(at least by visual inspection) surfaces. A patina normally refers to surface alteration by a

number of processes that result on modification of the upper layer, sometimes pure staining

(aesthetic) but also chemical transformation of the upper layer. Please clarify this point.

Answer 5. The Reviewer is right: The term “white” could be confusing. We collected this

powder from apparently not colonized area and not from “white patina”. We modified the

legend of the samples changing “white” in “uncolonized” area and we added a new table

with the ID of the samples and corresponding DNA extraction values (Table S1).

Question 6. Also, please state if they were taken randomly. In addition, indicate how you managed

with graffiti that is obvious on the image. Adding arrows to the sampled areas would have the reader

to clearly identity the origin of samples.

Answer 6. The materials and methods section was clarified as requested and arrows were

added in the figure.

Question 7. The apparent lack of correspondence of certain organisms not identified by NGS but

seen by microscopy could be explained by the non-efficient extraction of nucleic acids, we have

seen this in the past with thick-sheathed cyanobacteria.

Answer 7. We welcomed the comment of the Reviewer and we agree that difficulties in lysis

of specific bacterial sheaths and consequently DNA extraction could be responsible for the

not complete correspondence with the microscopic analyses results. For this reason we

added a specific note concerning this point in the revised manuscript.

Question 8. Overall, I find that the results need be better contrasted with published studies based

on both non-culture dependent and culture-dependent studies from epilithic habitats and highlight

the main findings.

Answer 8. The text was modified as requested.

Question 9. Please also state and provide relevant references as to how this type of NGS-based

study can provide relevant information regarding conservation issues.

Answer 9. We would like to emphasize that this is the first study that apply NGS to the

characterization of black patinas, so no references are available about the relevance of this

technique in the field of conservation. The importance of the obtained data has been

underlined in the text.

Question 10. In addition, the conclusion section needs to be enhanced to fully be supported by the

results.

Answer 10. The conclusion section was extensively revised and we think that now it’s better

supported by the results.

Reviewer # 2

General comment

The authors present an interesting study by analyzing the microbiota of a black patina often found

over travertine embankments of Tiber river in Rome. For this reason, Next-Generation sequencing

techniques, through Illumina platform, were applied in order to identify and characterize different

communities of bacteria, fungi and algae, as a mean to understand the possible effects of these

colonial organisms to the studied material. The study is well organized, presents relevant data,

especially the statistics results and graphics, and the manuscript is well written.

We have been very happy to learn that the Reviewer 2 found the MS “an interesting study”,

“well organized”, that “presents relevant data” and “well written”.

Specific points

Question 1. Introduction. Line 48 – replace “works of art” for “artworks”

Answer 1. The text was modified as indicated.

Question 2. In this section a final paragraph with the study objectives is missing. Please add the

objectives of the work to complete well the Introduction section.

Answer 2. The introduction was improved and the objectives of the work were added

Question 3. Materials and Methods. Line 103 – replace the number “1” for number “2”. In this

manuscript “Materials and Methods” are the section number 2.

Answer 3. The text was modified as indicated.

Question 4. Results. In sub-sections “3.1 Bacterial Community” and “3.2 Fungal community” please

add the percentages of abundance of the described taxa. This is relevant data in such NGS study

and is missing on this section.

Answer 4. We thank the Reviewer 2 for the suggestion. Two tables (one for bacteria and

one for fungi) containing the percentages of abundance for each taxon (averages and

standard deviations) in black patinas and uncolonized stone have been produced as

supplementary inserts.

Question 5. Discussion. I advise the authors to add an introductory paragraph to this section, instead

of starting immediately with the results discussion. It would be good to start with some statement (3-

4 lines) regarding the importance of the used methodology to characterize the microbiota

communities of the black patina present in such important Cultural Heritage structure, which was

actually the aim of this study.

Answer 5. We agree with the Reviewer 2 and an introductory statement was added to the

paragraph.

Reviewer # 3

General comment

I believe that the manuscript by Antonelli et al. “Tiber’s embankments black patina characterization

by Next-Generation Sequencing” (ref: PONE-D-19-21352) is a very interesting study concerning the

complete metagenomic analysis of black patinas in an important stone monument. In my opinion,

the topic is relevant and deserves to be highlighted. I also would like to pinpoint that the application

of Shotgun metagenomics is currently rather scarce in the field, thus turning the article highly

innovative. I recommend the acceptance of the article after the authors conduct major revisions in

the manuscript, and some crucial points are addressed.

We would like to thank the Reviewer 3 for considering our study “highly innovative” and for

his/her comments/suggestions that prompted us to improve the quality of the MS.

Specific points

Question 1. The article should be proofread by an English native speaker.

Answer 1. The manuscript has been proofread by an English native speaker

Question 2. The term 18S ITS should be replaced for ITS2-rDNA sub-region, since from my

understanding the 18S region (SSU) was not considered during the course of this study.

Answer 2. The word "18S ITS" has been replaced with " ITS2-rDNA " in the revised

manuscript

Question 3. Introduction section, Lines 49-60 and Lines 75-76.

Comment: Please consider rephrasing these sentences. They are too long and their structure could

be improved.

Answer 3. The text was modified as suggested.

Question 4. Line 71 and Line 284.

Comment: In line 71. the reference for Pentecost (1992), in this case [45], is missing. In line 284, the

reference for Albertano (2012), in this case [18], is also missing. I advise the authors to double check

their references along the manuscript and in the references.

Answer 4. The manuscript was checked and all the missing references were added.

Question 5. Introduction section, Line 64-84.

Comment: This part of the introduction section is only focused in Phototrophic microorganisms and

Fungi. I believe that the role (if any) and presence of bacteria in black patinas (if previously studied),

should also be highlighted in this part.

Answer 5. We agree with the Reviewer and, although the role of the bacteria in subaerial

biofilms had not been clarified jet, we added a section in the introduction regarding these

microorganisms.

Question 6. Introduction section, Line 93-95.

Comment: The aims of the study should be clearer.

Answer 6. The introduction was modified and the aims of the study were clarified

Question 7. Introduction section, Line 95-99.

Comment: I believe this part should be moved to the Materials and methods- Sample collection and

description sub-section.

Answer 7. The text was modified as indicated.

Question 8. Materials and methods section, Line 103. Discussion section, Line 274.

Comment: In line 103, this section should be: 2. Materials and methods. In the current form is

displayed as 1. Materials and methods. In line 274, this section should be: 4. Discussion. In the

current form lacks numbering.

Answer 8. The text was modified and the paragraph numbers were removed as requested

by the Plos One author’s guidelines

Question 9. Materials and methods section; 2.1 Sample collection and description.

Comment: I believe that the manuscript could benefit from the addition of a table displaying the

distinct samples IDs and further metadata. The table could also display which samples were able to

be studied through the distinct metagenomic methodologies applied.

Answer 9. Reviewer 3 is right. We added a table (Table S1), displaying the distinct sample

IDs, specific amount of DNA extraction, corresponding obtained library and sequencing

quality checkpoint.

Question 10. Materials and methods section; 2.1 Sample collection and description.

Comment: For the microscopical analysis, were the samples randomly selected? What were the

criteria for the selection of these samples? Which samples (ID) were studied?

Answer 10. We apologize if the description of this part of the material and methods section

was not clear. The section “Sample collection and description” was revised and now we

believe that we address the specific concerns regarding the selection of the samples.

Question 11. Materials and methods section; 2.2 DNA extraction, library preparation and

sequencing.

Comment: Please provide further details regarding the DNA extraction, library preparation and

sequencing.

Answer 11. The section “DNA extraction” in material and methods was added with more

detailed information, whereas the sections “Sample collection and description” and

“Sequencing Data Analysis” were revised.

Question 12. Results section; 3.2 Bacterial community, Lines 204-208.

Comment: This part should be moved to the discussion section.

Answer 12. The text was modified as suggested.

Question 13. Results (e.g. Lines 227-230), Discussion (e.g. Lines 341-383) and Figures 4 and 8.

Comment: These parts highlight my main concern with the manuscript. In general, the application of

the Illumina MiSeq methodology targeting the ITS2 rDNA sub-region does not allow a proper and

accurate taxonomic annotation to the species level. I believe that these parts as well as the figures

above mentioned, need to highlight a taxonomic annotation at the genus level, and therefore require

to be updated. I don´t feel that the discussion bulk will be affected by this decision. However, I do

acknowledge that this change will require several parts of the manuscript to be updated.

Answer 13. The figures were modified as requested and the Results and discussion

sections were updated

Question 14. Figure 6 legend needs further information, namely the distinct COG categories.

Answer 14. The revised version of Figure 6 contains the descriptions for each COG

category.

Question 15. Discussion section.

Comment: Given the quality of the results, some parts of the discussion section could be deeper

debated. For example, the authors state that some bacteria taxa can be linked to the presence of

substances of anthropogenic origin. Pollution is known to act as an ecological pressure on

cyanobacterial communities. Could the presence of these cyanobacterial taxa be linked to

extremotolerance profiles allowing them to thrive in these conditions? MCF are also known for their

tolerance to various environmental factors, could their presence in the B samples also be explained

by their high metabolic capacities?

Answer 15. The discussions were modified as requested. Two paragraphs concerning the

cellular and metabolic peculiarities of cyanobacteria and MCF were added

---

## [Decision Letter · Decision Letter 1]

18 Dec 2019

PONE-D-19-21352R1

Characterization of black patina from the Tiber River embankments using Next-Generation Sequencing

PLOS ONE

Dear Dr Guerrieri,

Thank you for submitting your manuscript to PLOS ONE. After careful consideration, we feel that it has merit but does not fully meet PLOS ONE’s publication criteria as it currently stands. Therefore, we invite you to submit a revised version of the manuscript that addresses the points raised during the review process.

Although the authors addressed most of the previous reviewers’ comments there are still some minor issues that need to be addressed, in order to fulfill the criteria to publish in PLOS ONE. In particular, improve the manuscript keywords, avoid redundant information. The second aim described in the introduction does not need a new paragraph, include it in the previous paragraph. Please also indicate the place/laboratory where the Illumina sequencing was performed (in line 165). The sentence given in lines 374-376 “The fact that these microorganisms have never before been reported in black patinas on stone artifacts is most likely due to the techniques routinely used in the field of cultural heritage, which are often not appropriate for their detection or identification.” must be properly supported, add a reference please.

Minor issues:

Include the sentences in lines 208-209 and 210-211 in the previous paragraph;

Line 140, Please replace “Twenty samples of black patina (B) and twenty controls (U) were used..” by “Twenty samples of black patina (B) and twenty controls (uncolonized region, U) were used…”;

Figure 3 and 4, include the color representing each sample in the PCoA biplot;

Figure 5 the species identified should be in italic;

Improve the graphics presented in figure 7 and 8, avoid the background colors;

The sentences from line 491-494 could be included in the previous paragraph. Moreover the sentence must be improved, it is not clear what the authors are reporting.

We would appreciate receiving your revised manuscript by Feb 01 2020 11:59PM. To enhance the reproducibility of your results, we recommend that if applicable you deposit your laboratory protocols in protocols.io, where a protocol can be assigned its own identifier (DOI) such that it can be cited independently in the future. For instructions see: http://journals.plos.org/plosone/s/submission-guidelines#loc-laboratory-protocols

We look forward to receiving your revised manuscript.

Kind regards,

Ana R. Lopes, PhD

Academic Editor

PLOS ONE

Reviewers' comments:

Reviewer's Responses to Questions

**Comments to the Author**

1. If the authors have adequately addressed your comments raised in a previous round of review and you feel that this manuscript is now acceptable for publication, you may indicate that here to bypass the “Comments to the Author” section, enter your conflict of interest statement in the “Confidential to Editor” section, and submit your "Accept" recommendation.

Reviewer #2: All comments have been addressed

Reviewer #3: All comments have been addressed

2. Is the manuscript technically sound, and do the data support the conclusions?

Reviewer #2: Yes

Reviewer #3: Yes

3. Has the statistical analysis been performed appropriately and rigorously? 

Reviewer #2: Yes

Reviewer #3: Yes

4. Have the authors made all data underlying the findings in their manuscript fully available?

Reviewer #2: Yes

Reviewer #3: Yes

5. Is the manuscript presented in an intelligible fashion and written in standard English?

Reviewer #2: Yes

Reviewer #3: Yes

6. Review Comments to the Author

Reviewer #2: The authors have made several improvements to the manuscript and I believe it is now ready for publication. All suggested comments have been fully addressed, as well as other important issues have been corrected and improved. I have no further comments on this work.

Reviewer #3: (No Response)

7. PLOS authors have the option to publish the peer review history of their article (what does this mean?). If published, this will include your full peer review and any attached files.

Reviewer #2: No

Reviewer #3: No

---

## [Author Response · Author response to Decision Letter 1]

20 Dec 2019

Object: Resubmission Manuscript PONE-D-19-21352R1

Dear Editor Ana R. Lopes,

We thank you for accepting with minor revision our manuscript entitled “Characterization of black patina from the Tiber River embankments using Next-Generation Sequencing” to PLOS ONE.

We feel that we have addressed all the minor issues and in particular:

- The manuscript keywords are now improved 

- The laboratory where the Illumina sequencing was performed was added

- Two new references were added to support the conclusions (Sterflinger K, Piñar G. Microbial deterioration of cultural heritage and works of art - Tilting at windmills? Appl Microbiol Biotechnol. 2013;97: 9637–9646. doi:10.1007/s00253-013-5283-1;

Ricci S, De Leo F, Urzì C, Guerrieri F, Antonelli F. Advantages of a multidisciplinary approach in the study and the characterisation of black patinas. In: Macchia A, Masini N, La Russa MF, Prestileo F, editors. Dialogues in Cultural heritage, Books of Abstracts of the 6th YOCOCU Conference. Matera: YOCOCU, CNR – IBAM (Istituto per i Beni Archeologici e Monumentali); 2018. pp. 277–280)

- Figure 7 and 8 were re-drawn

The authors all concur with this submission and affirm that the material presented in this manuscript has not been previously reported and is not under consideration for publication elsewhere. Additionally, the authors have no conflicting financial interests to disclose.

Looking forward to hearing from you at your earliest convenience,

Sincerely,

Francesca Guerrieri

Federica Antonelli

---

## [Editor Report · Decision Letter 2]

26 Dec 2019

Characterization of black patina from the Tiber River embankments using Next-Generation Sequencing

PONE-D-19-21352R2

Dear Dr. Guerrieri,

We are pleased to inform you that your manuscript has been judged scientifically suitable for publication and will be formally accepted for publication once it complies with all outstanding technical requirements.

With kind regards,

Ana R. Lopes, PhD

Academic Editor

PLOS ONE
---

## [Editor Report · Acceptance letter]

27 Dec 2019

PONE-D-19-21352R2 

Characterization of black patina from the Tiber River embankments using Next-Generation Sequencing 

Dear Dr. Guerrieri:

I am pleased to inform you that your manuscript has been deemed suitable for publication in PLOS ONE. Congratulations! Your manuscript is now with our production department. 

With kind regards,

on behalf of

Dr. Ana R. Lopes 

Academic Editor

PLOS ONE